# Super-broadband on-chip continuous spectral translation unlocking coherent optical communications beyond conventional telecom bands

Deming Kong [1✉], Yong Liu[1], Zhengqi Ren[2], Yongmin Jung[2], Chanju Kim [1], Yong Chen[2], Natalie V. Wheeler[2], Marco N. Petrovich[2], Minhao Pu [1], Kresten Yvind [1], Michael Galili[1], Leif K. Oxenløwe [1], David J. Richardson [2] & Hao Hu [1✉]

Today's optical communication systems are fast approaching their capacity limits in the conventional telecom bands. Opening up new wavelength bands is becoming an appealing solution to the capacity crunch. However, this ordinarily requires the development of optical transceivers for any new wavelength band, which is time-consuming and expensive. Here, we present an on-chip continuous spectral translation method that leverages existing commercial transceivers to unlock the vast and currently unused potential new wavelength bands. The spectral translators are continuous-wave laser pumped aluminum gallium arsenide on insulator (AlGaAsOI) nanowaveguides that provide a continuous conversion bandwidth over an octave. We demonstrate coherent transmission in the 2-μm band using well-developed conventional C-band transmitters and coherent receivers, as an example of the potential of the spectral translators that could also unlock communications at other wavelength bands. We demonstrate 318.25-Gbit s$^{-1}$ Nyquist wavelength-division multiplexed coherent transmission over a 1.15-km hollow-core fibre using this approach. Our demonstration paves the way for transmitting, detecting, and processing signals at wavelength bands beyond the capability of today's devices.

---

[1] DTU Fotonik, Technical University of Denmark, DK-2800 Kgs. Lyngby, Denmark. [2] Optoelectronics Research Centre, University of Southampton, Southampton SO17 1BJ, UK. ✉email: dmkon@fotonik.dtu.dk; huhao@fotonik.dtu.dk

Coherent optical communication is a truly revolutionary technology that has transformed old and overburdened optical fibre networks into data superhighways. Owing to its overwhelming advantages in terms of spectral efficiency and receiver sensitivity[1], coherent optical communication has significantly increased the capacity and transmission distance of fibre networks and has been critical to meeting the ever-increasing data traffic demands of the worldwide internet. Today, coherent technology has expanded its applications from long-haul, large-capacity transmission to metro networks and short-reach links[2] due to advances in electronic and photonic devices which have enabled tremendous cost and power consumption reductions[3]. Using coherent optical communication, information can be encoded onto optical carriers with advanced modulation[4,5] and multiplexing[6–8] in the complex domain by I/Q modulators. The complex field of the signal can be completely recovered after transmission by a coherent receiver, in which the linear transmission impairments such as chromatic dispersion can be fully compensated by digital signal processing[9,10]. Yet, coherent transceivers need to be specially designed and are only available for the conventional C and L bands[11,12], as well as recently the S band[13]. Today, coherent optical communication systems in the conventional wavelength bands are reaching their capacity limits[14–16]. The potential to open up new wavelength bands is now attracting significant interest[17]. New fibre technologies have emerged which give rise to the possibility of transmitting optical signals for instance in the 2-μm band[18–21] and the 1-μm band[22–24]. Also, an ambitious vision of utilizing the huge bandwidth resources of free space could support a possible universal solution for future ubiquitous optical communications with unlimited sustainability[25]. However, it is generally very challenging to build a complete coherent optical communication system for wavelength bands away from C and L, due to the lack of telecom-grade, commercially available narrow-linewidth lasers, I/Q modulators, and coherent receivers, including for wavelength bands such as O, E, S or U, and beyond. Developing these crucial devices is a cost-intensive and time-consuming process and it has been a major obstacle to opening up new wavelength bands.

An alternative solution is the use of spectral translation as a method to provide a bridge between the telecom bands to any other wavelength band of interest for optical transmission. By translating the wavelength of the signal forth and back using degenerate four-wave mixing (FWM), one can effectively build coherent transmitters and receivers for new wavelength bands. Demonstrations based on silicon nanowaveguides have shown the feasibility of bridging the telecom bands and the 2-μm band[26,27], exhibiting a large conversion bandwidth of 820 nm[26]. However, the conversion efficiency is limited due to intrinsic material properties and two-photon absorption (TPA) induced nonlinear loss. Thus, picosecond pulsed lasers with a high peak power are required for the pump. The need for a pulsed pump adds complexity and severely limits the speed and spectral efficiency of the signal. Recently, a highly efficient 650-nm spectral translation based on a continuous-wave (CW) pump has been demonstrated using a silicon nitride ($Si_3N_4$) microring resonator[28]. However, the nature of the resonance gives a discrete translation band in frequency and a narrow resonance bandwidth, which limits the data rate and compromises the potential for opening up new wavelength bandwidth.

Here, we propose and demonstrate continuous (in both time and frequency) spectral translation for coherent optical communication in the 2-μm band as an example of the potential to unlock new wavelength bands utilizing conventional C-band transceivers. Some of the preliminary results of this work have been reported in ref. [29]. In this paper, we provide a significant number of new findings and extension of results, analysis, and

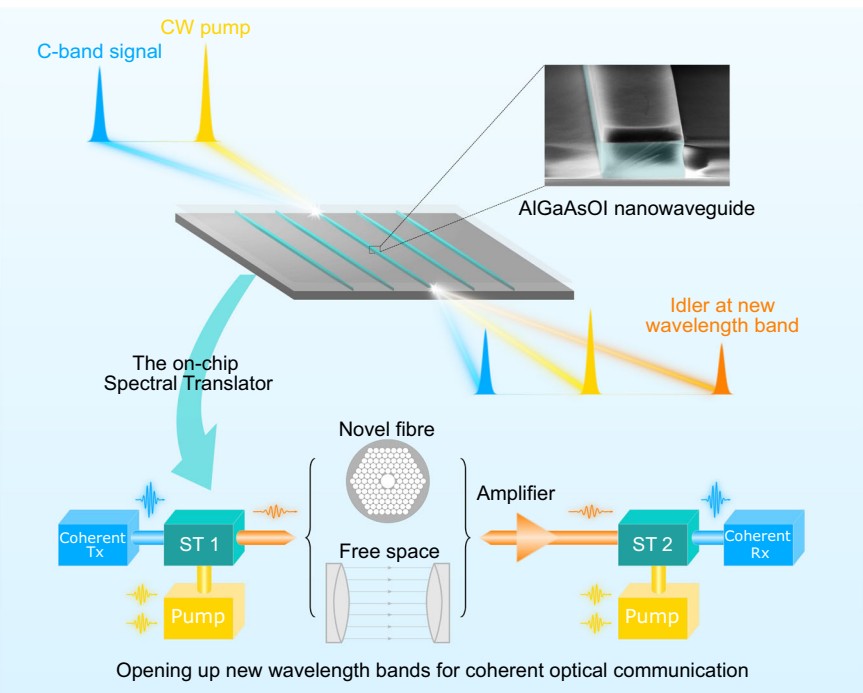

**Fig. 1 Opening up new wavelength bands for coherent optical communication through on-chip continuous spectral translation.** The spectral translators are based on continuous-wave (CW) pumped aluminium gallium arsenide on insulator (AlGaAsOI) nanowaveguides where a continuous conversion bandwidth over an octave can be achieved with reasonable conversion efficiency. Using a pair of spectral translators (ST 1 and ST 2), an optical coherent transmitter and a receiver working at new wavelength bands can be realized with high-performance C-band counterparts. Such a spectral translator-based coherent optical communication scheme has the potential to utilize the vast unexploited wavelength bands possible in novel fibres or free space without the need to develop optical transceivers at those wavelength bands.

discussion. Simulation shows our designed AlGaAsOI nanowaveguides could have a continuous translation band of over an octave. With the fabricated AlGaAsOI nanowaveguides as spectral translators, as well as a short-wavelength thulium-doped fibre amplifier (TDFA) and hollow-core fibre (HCF), we have enabled coherent transmission in the 2-µm band using C-band transmitters and coherent receivers. We have successfully transmitted 4 × 32-Gbaud 16 quadrature-amplitude modulation (QAM) Nyquist wavelength-division multiplexing (N-WDM) signals over a 1.15-km HCF at the 2-µm wavelength band with a spectral efficiency of 2.4 bit s$^{-1}$ Hz$^{-1}$. Beyond the application in optical transmission, the spectral translators may also find significance in a wider range of applications such as massive photonic integration in TPA-free wavelength bands, high-resolution spectroscopy for uncharted wavelength bands, as well as high-sensitivity infrared astronomy, due to the opportunities they provide for processing signals in wavelength bands that are currently inaccessible with today's devices.

## Results and discussion

**Continuous spectral translation opens up new wavelength bands for coherent optical communication.** As shown in Fig. 1, the basic principle of the proposed coherent optical communication at new wavelength bands is to bridge between the mature telecom 1.55-µm band (C band) and the new band by spectrally translating the signal forth and back using an on-chip solution based on CW-pumped degenerate FWM in AlGaAsOI nanowaveguides. We can then effectively build a pair of optical coherent transmitters and receivers that are compatible with advanced modulation and multiplexing technologies working in the new wavelength band based on mature C-band devices. At the transmitter side, advanced modulation and optical multiplexing such as QAM and N-WDM can be utilized and the high spectral efficiency signals converted to the new wavelength band. At the receiver side where the signal is converted back to the C band, coherent detection with advanced digital signal processing can be enabled for high performance and tolerance to linear impairments from the transceiver and transmission.

To achieve continuous spectral translation, both CW pumping and a non-resonant structure are needed, which puts a very strict requirement on the integrated nonlinear platform. Our work is based on recent advances in ultra-efficient AlGaAsOI nonlinear nanowaveguides[30]. The high refractive index contrast between the AlGaAs and the silica insulator can strongly confine light in the waveguide, enabling a very high conversion efficiency (CE) and bandwidth product amongst the various nonlinear material platforms[31]. The AlGaAsOI nanowaveguide can also be dispersion engineered to accommodate high-order phase matching for a very large conversion bandwidth of over an octave[32,33], supporting spectral translation with flexible target wavelength bands. The energy bandgap of AlGaAs can also be engineered by tuning the aluminium concentration, so that the nanowaveguide can be made free from TPA when pumping at the desired wavelength band, increasing the CE. These unique features make the AlGaAsOI nanowaveguide an appealing solution for continuous spectral translation.

Here, continuous spectral translation between the C band and the 2-µm band is explored and demonstrated, since the 2-µm band is becoming a leading contender for optical communication amongst new wavelength bands (Supplementary Note 2). Although we are targeting the 2-µm band in this demonstration, the spectral translation should work for any new potential wavelength band within the transparency window of the AlGaAsOI platform, i.e., from 500 nm to 10 µm[34]. Envisioning short-term applications, the proposed continuous spectral

translation can be used to activate the O-, E-, S- or U-bands in standard single-mode fibre (SSMF), thereby expanding the capacity of current telecommunication infrastructures. Optical communication in the 2-µm band, to the best of our knowledge, has focused on intensity modulation and direct detection[18–21,35]. Enabling a 2-µm-band coherent optical communication system based on spectral translation requires a sufficiently powerful 1.74-µm pump and AlGaAsOI nanowaveguides with a conversion bandwidth >450 nm. For the 1.74-µm pump, we have built an all-fibre TDFA working in the 1700–1800-nm band, achieving an output power of >500 mW[36]. The design and fabrication details of the short-wavelength TDFA are given in the 'Methods'.

**The AlGaAsOI nanowaveguides.** We have first explored the conversion bandwidth of the AlGaAsOI nanowaveguide using a numerical simulation. The geometric dimensions of the AlGaAsOI nanowaveguides are 910 nm in width, 350 nm in height and 5 mm in length. Figure 2a shows the simulation results of the dispersion profile of the AlGaAsOI nanowaveguide. The zero-dispersion wavelength (ZDW) is around 1747.7 nm. High-order phase matching can be used to expand the conversion bandwidth by combining both the fundamental phase matching band and high-order phase matching bands, albeit at some expense in CE[32,33]. Figure 2b illustrates the normalised phase mismatch $|\Delta\beta L/\pi|$ ($\Delta\beta$ is the phase mismatch and $L$ is the length of the waveguide, see Supplementary Note 1) as a function of pump and signal wavelengths, for a waveguide length of 5 mm. We see that a much wider continuous conversion bandwidth can be achieved by exploiting the fundamental and high-order phase matching bands, by slightly detuning the pump from the ZDW to the optimum wavelength at 1739.8 nm. Figure 2c shows the normalised CE (defined to characterize the wavelength-dependent conversion efficiency, see Supplementary Note 1) on a decibel scale against the length of the nanowaveguide and the signal wavelength when the pump is set to the optimum wavelength. The fundamental and high-order phase matching bands start to split when the length of the nanowaveguide increases. A synergetic conversion band is lost when the length surpasses the designed 5 mm. The conversion enables the continuous spectral translation from 1285.2 nm (233.27 THz) to 2691.9 nm (111.37 THz), spanning an octave. Figure 2d gives the normalised CE against the signal-idler separation in nanometres when the length of the nanowaveguide is set to 5 mm. Note that the conversion bandwidth is defined as the 3-dB width of the normalised CE against signal-idler separation. The overall conversion bandwidth is larger than 1400 nm with a flat (within 1 dB variation) conversion bandwidth of over 1000 nm. The design and fabrication details are given in the 'Methods'. The measured flatness of the translation band from 1500 to 1580 nm can be found in Supplementary Note 3. Spectral translation of an optical frequency comb (OFC) from the C band to the 2-µm band has been performed to explore the potential of our spectral translators (Supplementary Note 4).

**Experimental setup.** The continuous spectral translation is tested within the scenario of multidimensional modulation and multiplexing, in particular 32 Gbaud signal with 16-QAM format and four N-WDM channels. Figure 3a shows the experimental setup. We built up the 1.55-µm-band transmitter and receiver using discrete components in this demonstration for flexibility, but commercial transceivers should also work with the spectral translators. At the transmitter side, the 4 × 32 Gbaud 16-QAM signal is firstly generated using mature C-band external cavity lasers and I/Q modulators with a spacing of 33 GHz between adjacent channels. A CW single-mode distributed feedback

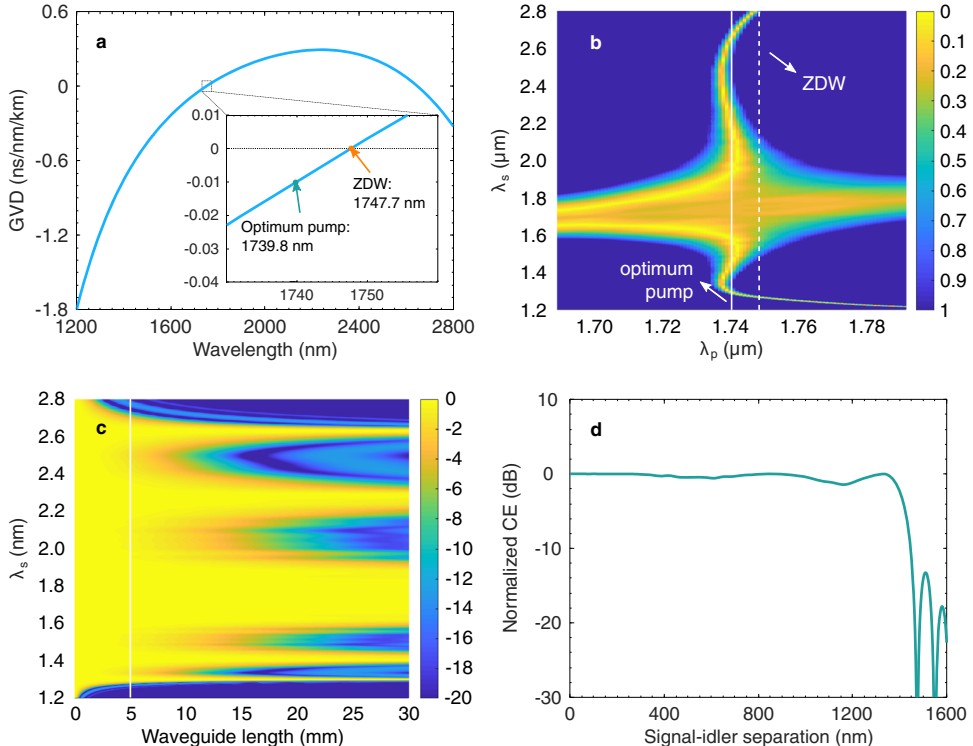

**Fig. 2 Simulation results of the AlGaAsOI nanowaveguide. a** Group velocity dispersion (GVD) of the AlGaAsOI nanowaveguide. The optimum pump wavelength (1739.8 nm) is shifted from the zero-dispersion wavelength (ZDW) (1747.7 nm) to utilize the high-order phase matching, where the fundamental conversion band can be merged with the high-order phase matching band for an extended conversion bandwidth. **b** Normalised phase mismatch $\left|\Delta\beta L/\pi\right|$ as a function of pump and signal wavelengths, with a waveguide length of 5 mm. The white straight solid and dash-dotted lines indicate the optimum pump wavelength and the ZDW. **c** Normalised CW as a function of waveguide length and signal wavelength with the pump at the optimum wavelength. **d** Normalised conversion efficiency (CE) versus signal-idler separation with a 5-mm-long AlGaAsOI nanowaveguide, when the pump is located at the optimum wavelength. The simulation results show a conversion bandwidth of over 1400 nm.

(DFB) laser is used as a pump, thermally tuned to a central wavelength of 1739.74 nm with a linewidth of 2 MHz and a power of 5 dBm. In order to simultaneously pump each nanowaveguide device (one at the transmitter and one at the receiver), and to maximize the CE, the CW pump was amplified by the short-wavelength TDFA and split into two with a 3-dB coupler, with measured optical powers of 20.0 and 19.9 dBm just before the input tapered couplers of AlGaAsOI nanowaveguides 1 and 2, respectively. The transmitter-side spectral translator converts the N-WDM signal to the 2-μm band. A 2-μm-band TDFA is used to amplify the translated N-WDM signal and reject the residual C-band signal and the pump.

We transmitted the 2-μm-band 4 × 32-Gbaud 16-QAM signal over a ~1.15-km low-latency hollow-core fibre (HCF, i.e., a 19-cell hollow-core photonic bandgap fibre[18] (see 'Methods' for further details)). The HCF has a flat-topped transmission window with the 3-dB width extending from 1959 to 2045 nm, with a minimum loss of 2.8 dB km$^{-1}$. It is spliced to two SSMF patch cords, giving an overall measured loss of 9 dB. The extra loss is mostly due to a large mode-field diameter mismatch between the HCF and the SSMF. The measured latency (i.e., pulse propagation delay) of the HCF is 3.82 μs (Supplementary Note 5).

At the receiver side, the 2-μm-band N-WDM signal is amplified and launched into the receiver-side spectral translator, where the 2-μm-band signal is converted back into the C band. An erbium-doped fibre amplifier (EDFA) amplifies the C-band N-WDM signal and rejects the residual 2-μm-band signal and the pump. The N-WDM signal is launched into a C-band coherent receiver for detection and performance evaluation. More details about the experimental setup, including the signal generation,

detection, and digital signal processing are described in the 'Methods'.

**System performance**. The optical spectra of the transmitter-side and receiver-side spectral translations are shown in Fig. 3b and c, respectively. The spectra are measured at the output of the AlGaAsOI nanowaveguides, thus the coupling losses are included. The transmitter-side spectral translator converts the 4 × 32-Gbaud 16-QAM signal centred at 1551.06 nm to a centre wavelength at 1980.58 nm. The transmitter-side CE is measured to be −35.4 dB. The receiver-side spectral translator converts the 2-μm-band signal back into the C-band with a centre wavelength of 1551.06 nm, with a measured CE of −24.5 dB. The difference in the CEs is mainly due to the larger coupling loss and waveguide loss experienced by the 2-μm-band signal than the C-band signal, as well as the difference between the AlGaAsOI nanowaveguides.

We evaluated the 2-μm-band transmission system by measuring the bit error ratio (BER) performance of a single 16-QAM channel, in the back-to-back scenario (Fig. 4a) and after 1.15-km HCF transmission (Fig. 4b). The received optical power is measured before the TDFA (TDFA 4 shown in Fig. 3) in the 2-μm-band coherent receiver. Since the 1.74-μm pump laser has a large linewidth of 2 MHz, we have observed severe cycle slips[37] when processing the received signal. Therefore, we used a 20% pilot overhead in the carrier recovery to remove the cycle slips. The pilot overhead could be significantly reduced if a carrier recovery algorithm with high tolerance to phase noise is used[38,39] or a narrow-linewidth laser is utilized for the 1.74-μm-band pump[40]. We have investigated the use of low-density parity-check (LDPC) code overheads of 20%, 25% and 33%. We assume a 0.8%

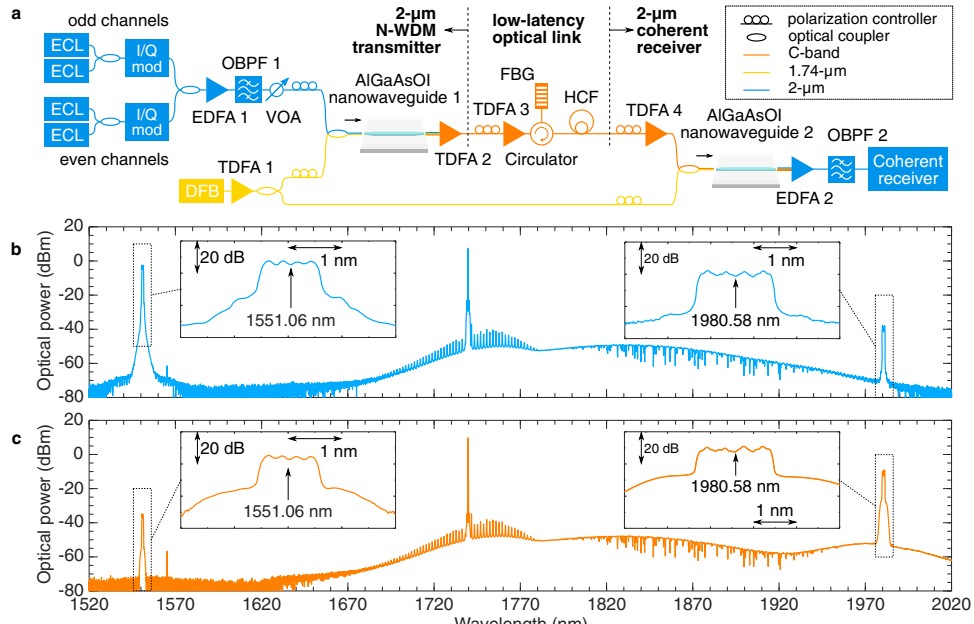

**Fig. 3 2-μm-band coherent transmission based on a pair of spectral translators with 4 × 32 Gbaud 16-QAM signal. a** The 2-μm-band N-WDM transmitter is based on a spectral translator (AlGaAsOI nanowaveguide 1) to convert the C-band signal to the 2-μm band. The 2-μm-band coherent receiver is based on a second spectral translator (AlGaAsOI nanowaveguide 2) to convert the 2-μm-band signal back into the C band. The pump for the spectral translators is generated through a distributed feedback (DFB) laser working at 1739.74 nm and a short-wavelength thulium-doped fibre amplifier (TDFA 1). The 2-μm-band signal is transmitted through a ~1.15-km hollow-core fibre (HCF) link with a low latency of 3.82 μs. **b** Transmitter-side spectral translation from 1551.06 to 1980.58 nm with an output CE of −35.4 dB. **c** Receiver-side spectral translation from 1980.58 nm back to 1551.06 nm with an output CE of −24.5 dB. The difference in the CE is due to the difference in the AlGaAsOI nanowaveguides and the fact that the 2-μm-band signal experiences more loss during coupling and propagation (ECL external cavity laser, EDFA erbium-doped fibre amplifier, OBPF optical bandpass filter, VOA variable optical attenuator, FBG fibre Bragg grating).

overhead for outer-hard-decision forward error correction (HD-FEC) to eliminate the well-known BER floor from the LDPC decoding[41]. We can conclude that all cases result in error-free performance after LDPC decoding. A ~2-dB penalty in terms of received optical power is observed after transmission. For the performance of the N-WDM signal (Fig. 4c, back-to-back; Fig. 4d, after transmission), we use 33% overhead for the LDPC coding to adapt to the signal-to-noise ratio of the received signal and achieve error-free performance. The line rate of the 2-μm-band 4 × 32-Gbaud 16-QAM signal is 32 Gbaud × 4 (16 QAM modulation) × 4 (N-WDM) = 512 Gbit s$^{-1}$. The net rate is 512 Gbit s$^{-1}$ ÷ (1 + 20%) ÷ (1 + 33%) ÷ (1 + 0.8%) = 318.25 Gbit s$^{-1}$, excluding the forward error correction (FEC) overheads and the pilot overhead. Since the spectral occupation of the N-WDM signal is only 131.32 GHz, the spectral efficiency of 2.4 bit s$^{-1}$ Hz$^{-1}$ is achieved. We have observed a 4.5-dB power penalty for the N-WDM signal after transmission. The penalties observed for the single-channel and N-WDM transmissions mainly come from the losses of the HCF and the receiver-side spectral translator. To improve the conversion efficiencies of the spectral translations, the transmitter-side and receiver-side AlGaAsOI nanowaveguides should be further optimized for the 2-μm-band and C-band signals, respectively, to minimize the waveguide loss and coupling loss.

We have proposed and demonstrated a scheme to unlock new wavelength bands for optical coherent communication using a pair of integrated continuous spectral translators based on CW pumped AlGaAsOI nanowaveguides and high-performance C-band transmitters and receivers. With a short-wavelength TDFA and HCF, we have demonstrated the spectrally translated 2-μm-band coherent optical transmission of high-capacity N-WDM signals with a spectral efficiency of 2.4 bit s$^{-1}$ Hz$^{-1}$. The spectral

translation scheme could be adapted to other wavelength bands, since the AlGaAsOI nanowaveguides can unlock at least an octave-spanning conversion bandwidth. This approach could facilitate the use of new wavelength bands for coherent optical communications still utilizing conventional C-band transceivers. We also believe this scheme has significance in broader applications beyond optical communications, such as in massive photonic integration in TPA-free wavelength bands, in high-resolution multi-banded spectroscopy and in high-sensitivity infrared astronomy, through the possibility of coherently transmitting, detecting and processing signals at wavelength bands that are currently inaccessible using current signal processing devices.

## Methods

**The design and fabrication of the AlGaAsOI nanowaveguide.** Each AlGaAsOI nanowaveguide used in the experiment consists of a straight waveguide and two inverse tapers serving as input and output ports to the waveguide. The core of the nanowaveguide has a high refractive index contrast with the silica cladding from both the top and bottom, leading to strong light confinement[30]. Thus, the required pump power for the spectral translation is significantly reduced. Dispersion engineering is efficiently achieved by adjusting the dimensions (height and width) of the nanowaveguide thanks to the highly confined waveguide mode, enabling a large conversion bandwidth. The high-order phase-matching technique is also used to extend the conversion bandwidth over 1400 nm[32]. The cross-section dimensions of the straight waveguide are designed to be 910 × 350 nm$^2$, resulting in a ZDW at 1747.7 nm. A pair of inverse tapers are placed at each side of the straight waveguide, and are optimized for low coupling loss. The nanowaveguide is 5 mm in length to enable a suitable balance between the conversion bandwidth and efficiency. Although the 910-nm-wide nanowaveguide supports higher-order modes besides the fundamental, the excitation of the higher-order modes is suppressed by the adiabatic design of the inverse taper and fully-straight design of the nanowaveguide.

The fabrication of the AlGaAsOI nanowaveguides starts with a GaAs substrate with a 3-μm-thick layer of silicon dioxide and a 350-nm-thick layer of GaAs on

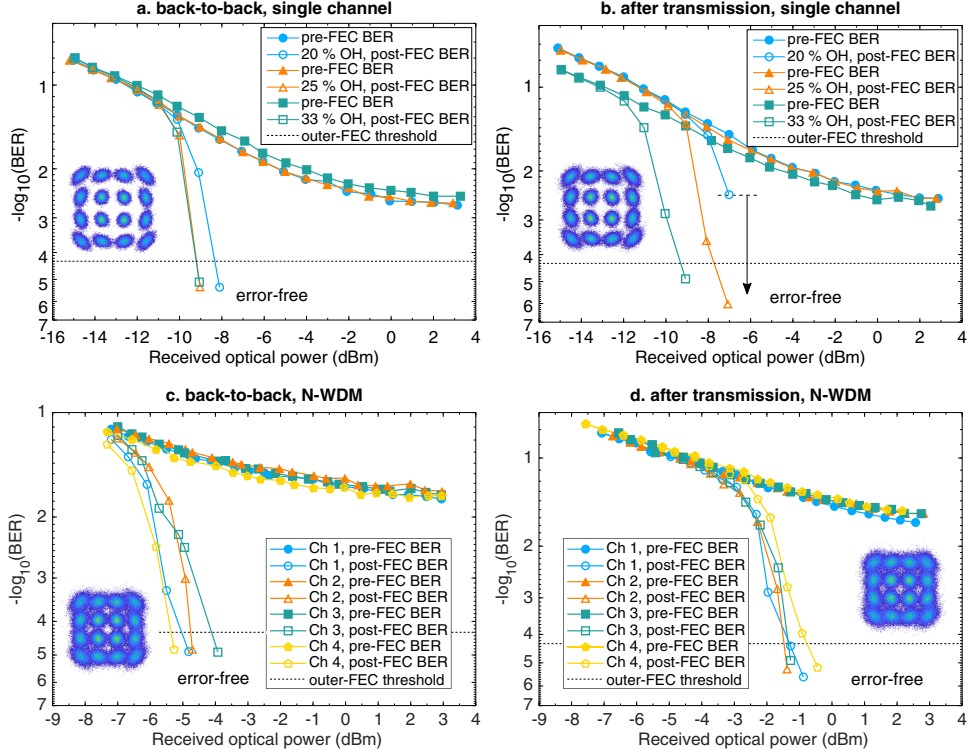

**Fig. 4 Bit error ratio (BER) measurement for the 2-μm-band signals using the 2-μm-band transmitter and receiver based on the spectral translators.**
**a** BER performance of the single-channel 2-μm-band 16-QAM signal at back-to-back. **b** BER performance of the single-channel 2-μm-band 16-QAM signal after 1.15-km HCF transmission. Error-free performance is achieved for both back-to-back and transmission scenarios with 20%, 25% and 33% LDPC overheads, indicating a good performance margin. **c** BER performance of the 2-μm-band 4 × 32 Gbaud Nyquist wavelength-division multiplexing (N-WDM) 16-QAM signal at back-to-back. **d** BER performance of the 2-μm-band 4 × 32 Gbaud N-WDM 16-QAM signal after 1.15-km HCF transmission. With 33% LDPC overhead, error-free performance is achieved for both back-to-back and transmission scenarios. Insets give the typical constellation diagrams of the 16-QAM signals that can achieve error-free performance after forward error correction (FEC) decoding.

top. The aluminium composition (x) for $Al_xGa_{1−x}As$ is 21%. The AlGaAsOI wafer was achieved by wafer bonding and substrate removal[42]. Then, the nanowaveguides and tapers were defined by E-beam lithography followed by a boron trichloride-based dry etching process. Both the E-beam process and the dry etching process were optimized to ensure minimal sidewall roughness[31]. The oxide-like E-beam resist hydrogen silsesquioxane was left on top of the nanowaveguide for the simplicity of fabrication. Another 3-μm-thick silicon dioxide layer was lastly deposited on top of the nanowaveguides by plasma-enhanced chemical vapour deposition to form a cladding. The sample was lastly cleaved at the inverse tapers, which were probed by lensed fibres for light coupling.

**The 1.74-μm-band TDFA.** For the 1.74-μm pump source, we developed an all-fibre short-wavelength (1650–1800 nm) TDFA using commercially available thulium-doped fibre (TDF) (OFS TmDF200). To enhance the short-wavelength gain of the TDFA a relatively short overall length of TDF (~1.2 m) was used to reduce the signal reabsorption and an additional short-pass spectral filter was incorporated within the amplifier to suppress gain and the build-up of amplified spontaneous emission/spurious lasing at longer wavelengths. The short-pass filter was based on macrobend-induced loss from a 10 m length of dispersion compensating fibre (DCF) (Thorlabs DCF38), wound on a mandrel at an optimized bend diameter of 5.5 cm. This filter was spliced into the middle of the active fibre to increase the available gain at the 1.74-μm band[36]. The fibre was core pumped in a bidirectional configuration by an in-house built erbium-doped fibre laser operating at 1560 nm. The amplifier provides a maximum small-signal gain of >30 dB at 1750 nm and supports gain over a more than 200 nm wide spectral window in the range 1730–1950 nm. In our experiment, the amplifier was seeded by a discrete mode DFB laser diode operating at 1739.74 nm (Eblana Photonics EP1742-DM-B) and the maximum achievable output power was higher than 500 mW with an optical signal-to-noise ratio (OSNR) of >40 dB (as shown in Fig. 3).

**The 1.15-km HCF.** The HCF used in our experiment is a hollow-core photonic bandgap fibre with a 19-cell core structure designed to operate in the 2 μm region and was previously fabricated in-house using a conventional stack and draw technique[20]. The central hollow core region is surrounded by multiple periodic layers of air holes and glass struts (i.e. micro-structured cladding with a $7\frac{1}{2}$-ring structure) and most of the light (>99%) propagates in air rather than silica glass

through the photonic bandgap effect. The core diameter is ~27 μm and the cladding diameter is ~166 μm (including the 105-μm diameter micro-structured region). The average hole-to-hole spacing is ~6 μm. The calculated mode field diameter is ~18 μm and the effective index of the fundamental mode is ~0.998. The fibre is 1.15-km long and shows a wide low loss transmission window centred around 2 μm, having a minimum loss of 2.8 dB km⁻¹ at 1993 nm and a 3 dB bandwidth of 86 nm (1959–2045 nm). Both ends of the HCF were spliced to conventional SSMF and the total link loss was ~9 dB. The link loss was dominated by loss due to the mode field diameter mismatch between the SSMF and HCF. Note that current state-of-the-art HCF now has a much lower attenuation (0.22 dB km⁻¹ now reported at wavelengths around 1625 nm)[43] and much lower interconnection losses can be achieved (0.15 dB now reported from SSMF to HCF)[44], and is emerging as an appealing solution for latency-sensitive optical interconnects.

**Data modulation and digital signal processing.** In the experiment, the C-band 4 × 32 Gbaud 16-QAM signal is generated by four external cavity lasers (ECLs) and two standard I/Q modulators. The ECLs, each with a linewidth of 10 kHz and output power of 15 dBm, are grouped into odd and even channels. The odd and even channels are then independently modulated by the I/Q modulators at 32 Gbaud with different source data. The data comes from a pseudo-random bit sequence with a length of $2^{23} − 1$. The data is encoded with LDPC block code from the digital video broadcasting-satellite-second generation (DVB-S2) standard and mapped to 16-QAM symbols. Then, the pilot symbols are added. The symbols are digitally shaped into waveforms using a root-raised-cosine filter with 401 taps and 0.01 roll-off. Finally, the waveforms are resampled and loaded to an arbitrary waveform generator with a sampling rate of 64 GSa s⁻¹ for the I/Q signals. The N-WDM signal is launched into the transmitter-side spectral translator with a measured optical power of 21 dBm before the AlGaAsOI nanowaveguide and then converted to the 2-μm band. A 2-μm-band TDFA is used as a link amplifier. A fibre-Bragg grating with a circulator as an optical bandpass filter (OBPF) is used to eliminate the excess amplified spontaneous emission noise.

The received 2-μm-band N-WDM signal is launched to the receiver-side spectral translator with a measured optical power of 15.7 dBm before the AlGaAsOI nanowaveguide and then converted back into the C-band. The C-band N-WDM signal is then amplified and filtered by a tuneable OBPF with 3-dB

bandwidth of 0.4 nm to pre-align a target N-WDM channel for performance investigation. The channel-under-test is then launched to the C-band coherent receiver, which consists of a polarization-diversity 90-degree hybrid, a tuneable local oscillator with 10-kHz linewidth and 16-dBm output power, four pairs of balanced photodetectors, and a digital sampling oscilloscope with 80 GSa s$^{-1}$ sampling rate and 33-GHz bandwidth. For each received power of each N-WDM channel, five 2-million-sample records from the oscilloscope are processed offline to evaluate the performance. The detected signals are digitally lowpass filtered and resampled to 2 samples per symbol. The signals are then synchronized and equalized using a T/2-spaced pilot-aided radius-directed adaptive equalizer with 51 taps. We use only 51 taps because the chromatic dispersion of the 1.15-km HCF can be neglected. The equalization rectifies inter-symbol interference due to the imperfect frequency response of the transmitter and receiver. A decision-directed phase-locked loop is applied for frequency offset correction and carrier phase recovery. Finally, the signal is LDPC decoded.

**Reporting summary**. Further information on research design is available in the Nature Research Reporting Summary linked to this article.

## Data availability

Source data are provided with this paper. The measurements data generated in this study have also been deposited in https://doi.org/10.5281/zenodo.6417777.

## Code availability

The algorithms used for the digital signal processing at the transmitter and the coherent receiver are standard and are outlined in detail in the 'Methods'. MATLAB scripts can be provided by the corresponding authors upon reasonable request.

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

## Acknowledgements

H.H. acknowledges the research grant (15401) of the Young Investigator Program (2MAC) from the VILLUM FONDEN. L.K.O. acknowledges funding from the Danish National

Research Foundation (DNRF) through the Research Centre of Excellence, Silicon Photonics for Optical Communications (SPOC) (ref. DNRF123). D.J.R. acknowledges the UK Engineering and Physical Sciences Research Council (EPSRC) through the "Airguide Photonics" Programme Grant (EP/P030181/1). N.V.W. acknowledges the Royal Society (University Research Fellowship). The authors gratefully acknowledge Francesco Poletti for his contribution to the design and fabrication of the 19-cell HCF.

## Author contributions

D.K. and H.H. conceived the concept and the experiment; Y.L. designed the AlGaAsOI nanowaveguides, supervised by H.H. and M.P.; Y.L. and D.K. performed the simulation of the AlGaAsOI nanowaveguides; Y.L., C.K. and K.Y. fabricated the AlGaAsOI nanowaveguides; D.K. and Y.L. characterized the AlGaAsOI nanowaveguides, and H.H. and M.P. provided suggestions; Z.R. and Y.J. designed and fabricated the 1.74-μm TDFA, supervised by D.J.R.; M.N.P., Y.C. and N.W. designed and fabricated the 19-cell HCF, supervised by D.J.R.; D.K. constructed the experiment setup, performed the experiment and processed the data; H.H. supervised the experiment; D.K., H.H., Y.L., Y.J., M.P., M.G., L.K.O. and D.J.R. discussed the results; the manuscript was written by D.K., Y.J. and H.H., and all authors contributed to the writing; H.H., L.K.O. and D.J.R. supervised the projects.

## Competing interests

The authors declare no competing interests.
