## [Peer Review File · Nature Communications]

Super-broadband On-chip Continuous Spectral Translation Unlocking Coherent Optical Communications Beyond Conventional Telecom BandsREVIEWER COMMENTS

Reviewer #1 (Remarks to the Author):

The paper entitled "Activating Unconventional Wavelength Bands for Coherent Optical Communications by On-Chip Continuous Spectral Translation" demonstrate a 318.25 Gbit/s Nyquist WDM coherent transmission over 1.15km of HC-PBGF at 2 μ m wavelength, by utilising a wavelength converter based on Gallium Arsenide on Insulator (AlGaAsOI) nanowaveguides.

The paper presents a number of issues.

(a) The claim on the 1st coherent experiment at 2 microns is quite strong and would recommend revision. Firstly, the authors have already demonstrated exactly the same system elsewhere (see references below). Secondly, the coherent detection (i.e. photodetectors) was at the C-band, not at 2 microns, hence it is not, by default, coherent detection at 2 μ m. It is important that the authors understand that it has been nearly 10 years since the 1st paper on 2 μ m [ref 22], and a lot of research and commercial activity has been in place all over the world with new exciting devices available at this waveband, including hybrids, balanced detectors etc. A quick search online will find those.

Refs:

[1] (2021) 2- μ m-band Coherent Transmission of Nyquist-WDM 16-QAM Signal by On-chip Spectral Translation, Deming Kong, Yong Liu, Zhengqi Ren, Yongmin Jung, Chanju Kim, Yong Chen, Natalie Wheeler, Minhao Pu, Kresten Yvind, Michael Galili, Leif K Oxenløwe, David J Richardson, and Hao Hu. Conference on Lasers and Electro-Optics OSA Technical Digest (Optica Publishing Group, 2021), paper SF1C.1 https://doi.org/10.1364/CLEO_SI.2021.SF1C.1

[2] (2020) Generation and heterodyne detection of a 2- μ m-band 16-QAM signal based on inter-band wavelength conversion, Yong Liu, Deming Kong, Zhengqi Ren, Yongmin Jung, Minhao Pu, Kresten Yvind, Michael Galili, Leif K Oxenlowe, David J Richardson, and Hao Hu Conference on Lasers and Electro-Optics OSA Technical Digest (Optica Publishing Group, 2020), paper SF2L.1

(b) The authors also claim the difficulty in creating devices that will operate at "unconventional bands". It is an unfortunate use of words, because the terminology "unconventional" normally relates to standard single mode fibers, and hence S- or O-bands etc (i.e. away from C+L). The device proposed doesn't address those. In fact, some of the claims that devices/materials are not available at the unconventional bands at O, E or S is not quite correct, see the paper below shows the availability of PICs operating at a wide range of wavelengths:

Ref:

[3] C. Doerr, L. Chen, T. Nielsen, R. Aroca, L. Chen, M. Banaee, S. Azemati, G. McBrien, S. Y. Park, J. Geyer, B. Guan, B. Mikkelsen, C. Rasmussen, M. Givehchi, Z. Wang, B. Potsaid, H. C. Lee, E. Swanson, and J. G. Fujimoto, "O, E, S, C, and L Band Silicon Photonics Coherent Modulator/Receiver," in Optical Fiber Communication Conference Postdeadline Papers, OSA Technical Digest (online) (Optica Publishing Group, 2016), paper Th5C.4.

(c) The issue here is that the paper tries to generalize the concept of spectral translation. Note that the photonics community has been calling "spectral translation" as "wavelength conversion" for decades. The concept of wavelength converting from C-band to unconventional bands (including S), has extensively been shown in the last 3-4 years, with the paper below as an example.

Ref:

[4] T. Kato, S. Watanabe, T. Yamauchi, G. Nakagawa, H. Muranaka, Y. Tanaka, Y. Akiyama, and T. Hoshida, "Whole Band Wavelength Conversion for Wideband Transmission," in Optical Fiber Communication Conference (OFC) 2021, P. Dong, J. Kani, C. Xie, R. Casellas, C. Cole, and M. Li, eds., OSA Technical Digest (Optica Publishing Group, 2021), paper F1B.1.

Having said that, when discussing wavelength conversion from 1.5 to 2microns, the submission is

spot on, and addresses it accordingly. In fact, the authors probably have one of the best devices that enable such conversion $>400\text{nm}$. Unfortunately, this device and the conversion expansion has already been shown in a number of prior publications in the past, hence the novelty is not really clear.

Refs:

[5] D. Kong et al., "Wavelength Conversion of 10 Gbit/s Data from 2000 to 1255 nm using an AlGaAsOI Nanowaveguide and a Continuous-Wave Pump in the C Band," 2019 Optical Fiber Communications Conference and Exhibition (OFC), 2019, pp. 1-3.

[6] D. Kong, Y. Liu, Z. Ren, Y. Jung, M. Pu, K. Yvind, M. Galili, L. K. Oxenløwe, D. J. Richardson, and H. Hu, "Generation and Coherent Detection of 2- μm -band WDM-QPSK Signals by On-chip Spectral Translation," in Optical Fiber Communication Conference (OFC) 2020, OSA Technical Digest (Optica Publishing Group, 2020), paper M1I.4.

[7] Kong, D., Ren, Z., Jung, Y., Chen, Y., Wheeler, N., Galili, M., Oxenløwe, L. K., Richardson, D. J., & Hu, H. (2021). 100 Gbit/s PAM-16 Transmission in the 2- μm Band over a 1.15-km Hollow-Core Fiber. In Proceedings of 2021 Optical Fiber Communication Conference IEEE.

(d) The motivation needs re-thinking. The suggestion on "unconventional bands" gives the impression on the use of standard single mode fibers, as I mentioned in (b). To explore standard single mode fibers, it makes sense to move to S, E, O- bands etc, and 2 μm would be too challenging due to the high loss, and hence unfeasible.

So then one assumes that the goal is the use of novel fibers, which is unconventional anyhow... So the terminology on unconventional bands is strange, because one has now opened the potential to explore and design transmission at any desired band ! And then, in this case, the explanation of which fibers and the choice of fibers is insufficiently discussed. The co-authors have extensive experience (and citations within the paper) on a number of new fibers that could unlock very wideband communications, not only with HC-PBGF but also with NANF. Wideband demonstrations using such fibers has also been shown in the past by co-authors. It is really unclear why limit the demo then to such short HC-PBGF instead, which was also used in a number of references within the submission.

My suggestion to authors here is either to focus on studying the potential of this device (or modified versions of the device) to enable S, E, O simultaneous conversion for wideband communications in standard single mode fibers (and even for multi-core / SDM as well to increase capacity), or explore ultra-wideband in NANF expanding to all sort of wavelengths available there. Alternatively, showing photonic integrated circuits (nonlinear device + detectors, for example) would be a real advantage with potential commercial opportunities too.

Reviewer #2 (Remarks to the Author):

The article proposes an original experimental proof of concept for a technology for using conventional C-band transceivers for optical communications on other spectral regions. The proposed technology could have a major impact on the optical communications by enabling to adapting standard TRX to different transmission media or to different bands on the same medium. The article is very-well written and concepts are clearly proposed and explained. The experiment has been rigorously conducted and results are clearly shown and commented.

I propose the acceptance of the paper for publication and I have only a couple of minor comments.

- The experiment has been based on lab discrete transceivers, but from the presented results I do not see any opposition in the use of conventional commercial transceivers. I invite the authors to shorty comment on this possibility.

- The authors target the POC to the use of the proposed technology on an unconventional band useful for HCF or FSO tranmission, and this is perfectly fine for a mid-to-long-term application. It should be interesting to comment on the possibility of using the proposed technique also for the transmission bands of the SMF besides the C+L-band, and specifically for the S, E and U-band. Such usage would envision also a short-term application of the proposed technology. I invite the authors to shorty comment on this possibility.

Response to review reports

We thank the editor and the reviewers for their thoughtful comments and valuable suggestions toward improving our manuscript. In the following, we present below a point-by-point response to the Reviewers' comments, with our comments in blue coloured text.

Reviewer #1 (Remarks to the Author):

The paper entitled "Activating Unconventional Wavelength Bands for Coherent Optical Communications by On-Chip Continuous Spectral Translation" demonstrate a 318.25 Gbit/s Nyquist WDM coherent transmission over 1.15km of HC-PBGF at 2 μ m wavelength, by utilising a wavelength converter based on Gallium Arsenide on Insulator (AlGaAsOI) nanowaveguides. The paper presents a number of issues.

Reviewer's comment 1:

(a) The claim on the 1st coherent experiment at 2 microns is quite strong and would recommend revision. Firstly, the authors have already demonstrated exactly the same system elsewhere (see references below). Secondly, the coherent detection (i.e. photodetectors) was at the C-band, not at 2 microns, hence it is not, by default, coherent detection at 2 μ m. It is important that the authors understand that it has been nearly 10 years since the 1st paper on 2 μ m [ref 22], and a lot of research and commercial activity has been in place all over the world with new exciting devices available at this waveband, including hybrids, balanced detectors etc. A quick search online will find those. Refs:

- [1] (2021) 2- μ m-band Coherent Transmission of Nyquist-WDM 16-QAM Signal by On-chip Spectral Translation, Deming Kong, Yong Liu, Zhengqi Ren, Yongmin Jung, Chanju Kim, Yong Chen, Natalie Wheeler, Minhao Pu, Kresten Yvind, Michael Galili, Leif K Oxenløwe, David J Richardson, and Hao Hu. Conference on Lasers and Electro-Optics OSA Technical Digest (Optica Publishing Group, 2021), paper SF1C.1
- [2] (2020) Generation and heterodyne detection of a 2- μ m-band 16-QAM signal based on inter-band wavelength conversion, Yong Liu, Deming Kong, Zhengqi Ren, Yongmin Jung, Minhao Pu, Kresten Yvind, Michael Galili, Leif K Oxenløwe, David J Richardson, and Hao Hu Conference on Lasers and Electro-Optics OSA Technical Digest (Optica Publishing Group, 2020), paper SF2L.1

Response:

First of all, the authors greatly appreciate the reviewer's valuable comments. The authors agree that we have presented part of our work in conference proceedings (Ref [1] listed above contains part of our work. Ref [2] listed above tested a heterodyne receiver and no transmission is conducted.) as pointed out by the reviewer but publishing work in conference proceedings before journal publication is common in research communities. Please note that our current submission provides a significant number of new findings and extension of results, analysis and discussion of implications well beyond our previous conference papers. For example, we have explored the

conversion bandwidth of the AlGaAsOI nanowaveguides through rigorous simulation/optimization and for the first time presented the unique octave spanning spectral translation over 1000 nm. Also, the other two key enabling components (i.e. high power thulium doped fibre amplifier at 1.74 μm and a 2 μm hollow core fibre) are described in depth in terms of fabrication method and characterization. More importantly, the topical editor of Nature Communications kindly confirms that they would not consider these conference proceedings as potentially compromising and they are thus interested in the possibility of publishing our work. However, it is worth incorporating a reference to our preliminary conference papers within the revised manuscript, two new sentences are added to page 3.

Modification in Page 3, Lines 67-68:

“Some of the preliminary results of this work have been reported in ref. ²⁹. In this paper, we provide a significant number of new findings and extension of results, analysis, and discussion.”

Regarding the second comment, the authors strongly believe that this is the first demonstration of a 2 μm -band coherent optical communication system using a spectral translation scheme without the use of 2- μm coherent transmitters/receivers. Our state-of-the-art AlGaAsOI nanowaveguides enable the use of new wavelength bands for coherent optical communication without the development of expensive transceivers/receivers and 2- μm -band Nyquist WDM signals have been successfully transmitted over a 1.15 km long hollow core fibre in our experiment. Therefore, this is the first 2- μm coherent transmission from the information transmission point of view and there is no need to use 2- μm transmitters and receivers for 2- μm transmission.

The authors agree that there has been much progress in the 2- μm region and various new and interesting devices have been developed, including photodiodes, balanced photodiodes, LiNbO₃ intensity modulators, TDFAs, etc. Yet, our point is, as shown in our manuscript from line 48 to line 49, the development of these devices for new potential transmission bands will take time and will be costly. If our technology (i.e., spectral translation) is utilized, it is possible to effectively transmit signals for any desired band within the transparency window of the AlGaAsOI nanowaveguide and this is much more time- and cost-effective approach compared to developing new transmitters and receivers from scratch. For the 2- μm band, we are aware that balanced photodetectors exist from Discovery Semiconductors but a practical solution for a full coherent receiver still remains elusive (including the narrow linewidth laser and the optical hybrid). To be more precise, we have modified our manuscript in page 2, line 43-45:

“However, it is generally very challenging to build up a complete coherent optical communication system for unconventional wavelength bands due to the lack of narrow-linewidth lasers, I/Q modulators, and coherent receivers.”

Reviewer's comment 2:

(b) The authors also claim the difficulty in creating devices that will operate at “unconventional bands”. It is an unfortunate use of words, because the terminology “unconventional” normally relates to standard single mode fibers, and hence S- or O-bands etc (i.e. away from C+L). The device proposed doesn't address those. In fact, some of the claims that devices/materials are not available at the unconventional bands at O, E or S is not quite correct, see the paper below shows the availability of PICs operating at a wide range of wavelengths:
Ref:

[3] C. Doerr, L. Chen, T. Nielsen, R. Aroca, L. Chen, M. Banaee, S. Azemati, G. McBrien, S. Y. Park, J. Geyer, B. Guan, B. Mikkelsen, C. Rasmussen, M. Givehchi, Z. Wang, B. Potsaid, H. C. Lee, E. Swanson, and J. G. Fujimoto, "O, E, S, C, and L Band Silicon Photonics Coherent Modulator/Receiver," in Optical Fiber Communication Conference Postdeadline Papers, OSA Technical Digest (online) (Optica Publishing Group, 2016), paper Th5C.4.

Response:

We thank the reviewer for pointing out the confusion in the terminology we used. As we described in the first four paragraphs of our manuscript, we meant that we could open up the possibility to utilize new wavelength bands by using spectral translations. We do not refer specifically to the S, O, or E band, as usually indicated for the transmission windows of standard single-mode fibres. Our continuous (in both time and frequency) spectral translations based on the AlGaAsOI nanowaveguides could provide an ultra-large conversion bandwidth that has simply not been shown in any other demonstrations. This is indeed the most important novelty of this work. In our revised manuscript, we have explicitly explained our target wavelength bands (page 2, line 45-48):

“The “unconventional” wavelength bands here do not refer to the O-, E-, S-, or U- bands that are traditionally associated with standard single-mode fibre (SSMF), but refer to any new potential wavelength band that supports optical transmission through optical fibre or free space.”

We have also commented on the possible working wavelengths of the AlGaAsOI material for the spectral translations and the possibility of applying the spectral translation to the O-, E-, S-, or U-bands (from page 4, line 105-107):

“Although we are targeting the 2- μm band in this demonstration, the spectral translation should work for any new potential wavelength band within the transparency window of the AlGaAsOI platform, i.e., from from 500 nm to 10 μm ³⁴, including the O-, E-, S-, or U- bands in the SSMF.”

Reviewer's comment 3:

(c) The issue here is that the paper tries to generalize the concept of spectral translation. Note that the photonics community has been calling "spectral translation" as "wavelength conversion" for decades. The concept of wavelength converting from C-band to unconventional bands (including

S), has extensively been shown in the last 3-4 years, with the paper below as an example.

Ref:

[4] T. Kato, S. Watanabe, T. Yamauchi, G. Nakagawa, H. Muranaka, Y. Tanaka, Y. Akiyama, and T. Hoshida, "Whole Band Wavelength Conversion for Wideband Transmission," in Optical Fiber Communication Conference (OFC) 2021, P. Dong, J. Kani, C. Xie, R. Casellas, C. Cole, and M. Li, eds., OSA Technical Digest (Optica Publishing Group, 2021), paper F1B.1.

Having said that, when discussing wavelength conversion from 1.5 to 2microns, the submission is spot on, and addresses it accordingly. In fact, the authors probably have one of the best devices that enable such conversion >400nm. Unfortunately, this device and the conversion expansion has already been shown in a number of prior publications in the past, hence the novelty is not really clear.

Refs:

[5] D. Kong et al., "Wavelength Conversion of 10 Gbit/s Data from 2000 to 1255 nm using an AlGaAsOI Nanowaveguide and a Continuous-Wave Pump in the C Band," 2019 Optical Fiber Communications Conference and Exhibition (OFC), 2019, pp. 1-3.

[6] D. Kong, Y. Liu, Z. Ren, Y. Jung, M. Pu, K. Yvind, M. Galili, L. K. Oxenløwe, D. J. Richardson, and H. Hu, "Generation and Coherent Detection of 2- μ m-band WDM-QPSK Signals by On-chip Spectral Translation," in Optical Fiber Communication Conference (OFC) 2020, OSA Technical Digest (Optica Publishing Group, 2020), paper M11.4.

[7] Kong, D., Ren, Z., Jung, Y., Chen, Y., Wheeler, N., Galili, M., Oxenløwe, L. K., Richardson, D. J., & Hu, H. (2021). 100 Gbit/s PAM-16 Transmission in the 2- μ m Band over a 1.15-km Hollow-Core Fiber. In Proceedings of 2021 Optical Fiber Communication Conference IEEE.

Response:

We thank the reviewer for pointing these out.

Unlike “wavelength conversion”, the terminology “spectral translation” usually refers to demonstrations from one wavelength band to another, with a bandwidth over hundreds of nanometers. The “spectral translation” terminology has been used in many publications, including references [26] and [28]. To distinguish our work from wavelength conversion demonstrations with a single channel and a few tens of nanometers, we strongly prefer to adopt the terminology “spectral translation” in our manuscript.

[26] Liu, X. et al. Bridging the mid-infrared-to-telecom gap with silicon nanophotonic spectral translation. *Nature Photonics* 6, 667–671 (2012).

[28] Lu, X. et al. Efficient telecom-to-visible spectral translation through ultralow power nonlinear nanophotonics. *Nature Photonics* 13, 593–601 (2019).

For our previous presentations at conferences, again, according to the journal’s policy, Nature Communications would not consider these conference proceeding as potentially compromising.

Reviewer’s comment 4:

(d) The motivation needs re-thinking. The suggestion on “unconventional bands” gives the impression on the use of standard single mode fibers, as I mentioned in (b). To explore standard single mode fibers, it makes sense to move to S, E, O- bands etc, and 2um would be too challenging due to the high loss, and hence unfeasible. So then one assumes that the goal is the use of novel fibers, which is unconventional anyhow... So the terminology on unconventional bands is strange, because one has now opened the potential to explore and design transmission at any desired band ! And then, in this case, the explanation of which fibers and the choice of fibers is insufficiently discussed. The co-authors have extensive experience (and citations within the paper) on a number of new fibers that could unlock very wideband communications, not only with HC-PBGF but also with NANF. Wideband demonstrations using such fibers has also been shown in the past by co-authors. It is really unclear why limit the demo then to such short HC-PBGF instead, which was also used in a number of references within the submission. My suggestion to authors here is either to focus on studying the potential of this device (or modified versions of the device) to enable S, E, O simultaneous conversion for wideband communications in standard single mode fibers (and even for multi-core / SDM as well to increase capacity), or explore ultra-wideband in NANF expanding to all sort of wavelengths available there. Alternatively, showing photonic integrated circuits (nonlinear device + detectors, for example) would be a real advantage with potential commercial opportunities too.

Response:

We thank the reviewer for clarifying the novelty of our work, and the suggestions for our future research.

We have explained the terminology “unconventional wavelength bands” in our response to the previous questions (Comment #3) and we do agree that we can potentially open up any desired band within the transparency window of the AlGaAsOI nanowaveguides using the spectral translation technology, which has been emphasized several times in our manuscript.

For the transmission medium, we have mentioned in the first paragraph of our manuscript that opening up new wavelength bands is not only beneficial to optical fibre transmissions (SSMF, HC-PBGF, NANF, etc.) but that it could also be used for free-space optical transmissions where a vast amount of wavelength resource is available. The length of the HC-PBGF used in our experiment is not crucial for the novelty. We are simply showing a proof-of-concept demonstration for a complete coherent transmission system, including a 2- μ m-band transmitter, receiver, and transmission fibre. From the BER results, we can confidently say that a longer transmission distance is possible, as a large margin for error-free performance (after LDPC decoding) exists.

Reviewer #2 (Remarks to the Author):

The article proposes an original experimental proof of concept for a technology for using conventional C-band transceivers for optical communications on other spectral regions. The proposed technology could have a major impact on the optical communications by enabling to adapting standard TRX to different transmission media or to different bands on the same medium. The article is very-well written and concepts are clearly proposed and explained. The experiment has been rigorously conducted and results are clearly shown and commented. I propose the acceptance of the paper for publication and I have only a couple of minor comments.

Reviewer's comment 1:

- The experiment has been based on lab discrete transceivers, but from the presented results I do not see any opposition in the use of conventional commercial transceivers. I invite the authors to shorty comment on this possibility.

Response:

We thank the reviewer for pointing out this possibility. There is no major obstacle to adapting conventional (1.55- μm) commercial transceivers for the spectral translation. In fact, those commercial transceivers usually perform better than transceivers built up from discrete devices. One can attach the spectral translators to the transmitter and receiver to enable the signal generation and detection at new transmission bands. The only issue is that these spectral translators will introduce some degradation to the signal quality, including OSNR degradation and additional phase noise due to the pump laser. We have pointed out these penalties in the analysis of our result. We believe there is still room for improvement in translation efficiency by optimising the AlGaAsOI nanowaveguides and the phase noise issue can be greatly mitigated if a narrow linewidth laser is used for the pump, or phase-noise tolerant carrier recovery algorithms are used.

We have added some comments on this in our revised manuscript (page 9, lines 141-143):

“We built up the 1.55- μm -band transmitter and receiver using discrete components in this demonstration for flexibility, but commercial transceivers should also work with the spectral translators without any obstacle.”

Reviewer's comment 2:

- The authors target the POC to the use of the proposed technology on an unconventional band useful for HCF or FSO transmission, and this is perfectly fine for a mid-to-long-term application.

It should be interesting to comment on the possibility of using the proposed technique also for the transmission bands of the SMF besides the C+L-band, and specifically for the S, E and U-band. Such usage would envision also a short-term application of the proposed technology. I invite the authors to shortly comment on this possibility.

Response:

We thank the reviewer for this suggestion. It is possible to convert from the C band to O, S, E, or U band using the AlGaAsOI nanowaveguides with some modifications to the design. And this is indeed very interesting as utilizing these bands in SMF has been a research focus recently. Some demonstrations have been shown in the literature for combined S-, C-, and L-band transmission, e.g., for SMF [R1, R2] and multicore fibres [R3].

In the revised manuscript, we have firstly distinguished the O-, E-, S-, and U- bands from the “unconventional bands” referred in the manuscript (page 2, 45-48):

“The “unconventional” wavelength bands here do not refer to the O-, E-, S-, or U- bands that are traditionally associated with standard single-mode fibre (SSMF), but refer to any new potential wavelength band that supports optical transmission through optical fibre or free space.”

Then, we have commented on the possibility of utilizing the O-, E-, S-, and U- bands (from page 4, 105-109):

“Although we are targeting the 2- μm band in this demonstration, the spectral translation should work for any new potential wavelength band within the transparency window of the AlGaAsOI platform, i.e., from 500 nm to 10 μm ³⁴, including the O-, E-, S-, or U- bands in the SSMF. Envisioning short-term applications, the proposed continuous spectral translation can be used to activate the O-, E-, S-, or U- bands in the SSMF, thereby expanding the capacity of current telecommunication infrastructures.”

[R1] J. Renaudier et al., “First 100-nm continuous-band WDM transmission system with 115Tb/s transport over 100 km using novel ultra-wideband semiconductor optical amplifiers,” in Proc. Eur. Conf. Opt. Commun., 2017, Art. no. Th.PDP.A.

[R2] T. Kato, S. Watanabe, T. Yamauchi, G. Nakagawa, H. Muranaka, Y. Tanaka, Y. Akiyama, and T. Hoshida, "Whole Band Wavelength Conversion for Wideband Transmission," in Optical Fiber Communication Conference (OFC) 2021, P. Dong, J. Kani, C. Xie, R. Casellas, C. Cole, and M. Li, eds., OSA Technical Digest (Optica Publishing Group, 2021), paper F1B.1.

[R3] B. J. Puttnam et al., "0.61 Pb/s S, C, and L-Band Transmission in a 125 μ m Diameter 4-Core Fiber Using a Single Wideband Comb Source," in *Journal of Lightwave Technology*, vol. 39, no. 4, pp. 1027-1032, 15 Feb.15, 2021, doi: 10.1109/JLT.2020.2990987.

REVIEWERS' COMMENTS

Reviewer #1 (Remarks to the Author):

Dear authors

Thanks for the revision. In my view, the key of the paper is the spectral translator (ok, I will keep this terminology!), and the potential in unlocking new wavelength bands for optical fiber and free space communications, as authors pointed out. It is an achievement of excellence. However, I still don't think the text is giving justice to the findings as it is. In particular, I would say, the motivation.

After reading the replies from the authors and looking into more details into the simulations, it is clear to me that the device to cater to other wavelength bands, but I am still uncertain if it is the same device or a modified-device (or what needs to be modified to be applied to other bands). And while I understand that this hasn't been done yet experimentally, I think you should focus on the fact that it could and will in the future. I see the 2microns experiments as an example of the potential of such device, not the end-game. Coherent is one aspect, there could be so many more.

So, as the editors don't mind the prior conference publications on this topic, and the fact that the spectral translator results should really be seen by a wider audience within Nature Comms, I am recommending a detailed review of the text, modifying some of the plain English expressions, and trying to really emphasize the real novelty proposed. I do hope you take them into account, so the target audience can truly appreciate the findings without the distraction of the specific target band, or detection method, or modulation used for the proof-of-concept.

The paper still needs a picture of the device itself that was used for this demonstration, with the typical semiconductor layer composition, dimensions, and how it was coupled to fibre (single mode? taper needed? Grating coupler?). I added specific comments on this below.

I still disagree with the terminology "unconventional bands". It's used in the text widely, but I will leave this Nat Comms editors to decide.

Detailed suggestions are:

1. Title. Re-phrase it to simply "Super-broadband on-chip continuous spectral translation unlocking coherent optical communications beyond S+C+L" or unlocking coherent communications at 2 microns.

2. Line 21: replace "This enables unprecedented coherent transmission in the 2- μ m band using well-developed conventional C-band transmitters and coherent receivers" with "this paper/letter shows a 700nm [or THz, whatever is the best number] spectral translator for coherent transmission in the 2- μ m band using well-developed conventional C-band transmitters and coherent receivers, as an example of this device potential, which could also unlock communications at other wavebands."

3. Lines 26-39 are not relevant, and not the best motivation for the technology. Better to start at line 39 at "The potential to open..."

4. Then line 44: I would replace with "However, it is generally very challenging to build a complete coherent optical communication system for wavelength bands away from C+L, due to the lack of telecom-grade, commercially available narrow-linewidth lasers, I/Q modulators, and coherent receivers, including for bands such as O-, E-, S- or U-, and beyond."

5. Line 50, replace: "An alternative solution is the use spectral translation technology to provide a bridge between mature telecom bands and unconventional wavelength bands" with "An alternative solution is the use of spectral translation [reference] as a means to provide a bridge between telecom bands to any other wavelength band, including 2microns in this case."

6. Line 59 onwards – very repetitive, including the paragraph that follows it. Replace with “However, the nature of the resonance gives a discrete translation band in frequency and a narrow resonance bandwidth, which limits the data rate and compromises the potential on opening up the new wavelength bandwidth.” I would also delete “Aiming for coherent optical communication with high speed and high spectral efficiency in unconventional wavelength bands, continuous (in both time and frequency) spectral translation is necessary but has not yet been demonstrated”.

7. Line 65, I would replace “Here, we propose and demonstrate a continuous spectral translation scheme for coherent optical communication at 2microns, as an example of the potential for unlocking the use of new wavelength windows utilizing conventional C-band transceivers. ”

8. Line 68 - Delete “We introduce the basic principle of continuous spectral translation and our AlGaAsOI platform for coherent optical communication in unconventional wavelength bands. We target the 2- μm wavelength band for the demonstration”. It’s not well written, it’s repetitive, and the previous sentences already explain that.

9. Line 72 onwards, replace with “With the fabricated AlGaAsOI nanowaveguides as spectral translators, as well as a short-wavelength thulium-doped fibre amplifier (TDFA) and hollow-core fibre (HCF), we have enabled coherent transmission in the 2- μm band using C-band transmitters and coherent receivers. We have successfully transmitted 4 \times 32-Gbaud 16-QAM Nyquist wavelength-division multiplexing (N-WDM) signals over a 1.15-km HCF at the 2- μm wavelength band with a spectral efficiency of 2.4 bit s⁻¹ Hz⁻¹”. The words “in-house”, or “unprecedented” are not appropriate. The matter of fact is how wide your spectral translation is, and everything else is an application of the device. TDFA and HCF developed by ORC that were used here were demonstrated before, and well referenced in this paper. Focus on what this paper is really trying to say. Perhaps here you should focus on operations of the spectral translator, “cw pumped at xxx, with yy bias, and at zzz oC” or whatever the actual operations details were – this gives a guide to the audience on the potential comparison with the nonlinear micro-ring resonators you mentioned before.

10. Line 79: the end is calling for the addition of other applications beyond comms, perhaps space applications, sensing, astronomy?! This draws further audiences to read your paper, and Nature-based publications tend to attract a wider community!

11. Lines 103-109 are excellent. I do like the new sentences there.

12. Line 109 onwards I would re-write as “Optical communication in the 2- μm band, to the best of our knowledge, has focused on intensity-modulation and direct-detection [refs]. Enabling a 2- μm -band coherent optical communication system based on spectral translation requires a sufficiently powerful 1.74- μm pump and AlGaAsOI nanowaveguides with a conversion bandwidth >450 nm. For the 1.74- μm pump, we have built an all-fibre TDFA working in the 1700-1800 nm band, achieving an output power of >500 mW [ref]. The design and fabrication details of the short-wavelength TDFA are given in the Methods.”

13. Line 116 – I still can’t see the fabrication details, dimensions and SEM picture of the AlGaAsOI nanowaveguide. Figure 1 has a representative picture, but it needs a new picture with the design (cross-section of the different layers / dimensions / compositions / refractive indexes), length etc. Furthermore, were tapers used? Or grating coupler? How was it coupled to fibre? Loss? It says that this information is in “Supplemental Information #3”, but I can’t find it in the reviewer pack. I can only see on “Methods”, and the figure is still missing.

14. Line 131 – this is the octave claim. Is this true for the same device? Or would different structures/composition be required to achieve the octave spectral translation?

15. Line 141 – rephrasing to “The 1.55- μm -band transmitter and receiver comprised of discrete components in this demonstration for flexibility, but commercial transceivers would equivalently work with the spectral translators”. Remove “without obstacles”, as it’s a fact, otherwise you need to explain what obstacles you were thinking about. So just delete it.

16. Line 145 – rephrasing to “A CW single mode [was it single mode?] distributed feedback (DFB) laser was used as a pump, thermally tuned to a central wavelength of 1739.74 nm with a linewidth of 2 MHz and a power of xx dBm. In order to simultaneously optically pump each nanowaveguide device (one at the transmitter and one at the receiver), and to maximize the CE, the CW pump was amplified by a short-wavelength TDFA and split into two with a 50/50 coupler (as it 50/50?), with measured optical powers of 20.0 dBm and 19.9 dBm just before launching to the input tapered couplers of each of the AlGaAsOI nanowaveguides (1 and 2), respectively.” Note: were the nanowaveguides fibre coupled? Or was it launched via free-space?

17. Figure 4: I must say that the colour code is unfortunate and the contrast is poor. I would imagine that people who have colour-reading issues would find difficulty in understanding the results. Always have in mind what it would look like if the figure was black and white, can you still see differences? My suggestion is to use very clear symbols (open/close) for pre- / post-compensation, use larger symbols etc. Hopefully the editors will work with you on them. Also confusing is the constellation diagrams. For what power level were they taken? An arrow to show that would be good.

18. Line 199 onwards (from “with a cutting-edge” etc. These are subjective wording, the use of plain English is paramount) - rephrase it to “With a short-wavelength TDFA and HCF, we have demonstrated the first spectrally translated 2- μ m-band coherent optical transmission of high capacity N-WDM signals with a spectral efficiency of 2.4 bit/s/Hz. The spectral translation scheme could be adapted to other wavelength bands, since the AlGaAsOI nanowaveguides can unlock at least an octave spanning conversion bandwidth. This approach could facilitate the use of new wavelength bands for coherent optical communications still utilizing conventional C-band transceivers and/or transponders. We also believe this scheme has significance in broader applications beyond optical communications, such as in massive photonic integration in TPA-free wavelength bands and in high-resolution multi-banded spectroscopy, through the possibility of coherently transmitting, detecting, and processing signals at the wavelength bands that are currently inaccessible using current signal processing devices.”

19. Line 210 - The design and fabrication of the AlGaAsOI nanowaveguide: It needs a picture with the details in here. See comment #13. I cannot see “supplemental information #3” in the reviewer pack, so I cannot comment on that.

Reviewer #2 (Remarks to the Author):

The Authors fully addressed my remarks that are properly clarified within the revised article. Therefore, my recommendation is for the article acceptance for publication.

Response to review reports

We thank the editor and the reviewers for their thoughtful comments and valuable suggestions towards improving our manuscript. Our point-by-point responses to the Reviewers' comments are given below, whereas our comments are marked in blue coloured text.

Reviewer #1 (Remarks to the Author):

Dear authors

Thanks for the revision. In my view, the key of the paper is the spectral translator (ok, I will keep this terminology!), and the potential in unlocking new wavelength bands for optical fiber and free space communications, as authors pointed out. It is an achievement of excellence. However, I still don't think the text is giving justice to the findings as it is. In particular, I would say, the motivation.

After reading the replies from the authors and looking into more details into the simulations, it is clear to me that the device to cater to other wavelength bands, but I am still uncertain if it is the same device or a modified-device (or what needs to be modified to be applied to other bands). And while I understand that this hasn't been done yet experimentally, I think you should focus on the fact that it could and will in the future. I see the 2 microns experiments as an example of the potential of such device, not the end-game. Coherent is one aspect, there could be so many more.

So, as the editors don't mind the prior conference publications on this topic, and the fact that the spectral translator results should really be seen by a wider audience within Nature Comms, I am recommending a detailed review of the text, modifying some of the plain English expressions, and trying to really emphasize the real novelty proposed. I do hope you take them into account, so the target audience can truly appreciate the findings without the distraction of the specific target band, or detection method, or modulation used for the proof-of-concept.

The paper still needs a picture of the device itself that was used for this demonstration, with the typical semiconductor layer composition, dimensions, and how it was coupled to fibre (single mode? taper needed? Grating coupler?). I added specific comments on this below.

I still disagree with the terminology "unconventional bands". It's used in the text widely, but I will leave this Nat Comms editors to decide.

Detailed suggestions are:

Reviewer's comment 1:

Title. Re-phrase it to simply "Super-broadband on-chip continuous spectral translation unlocking coherent optical communications beyond S+C+L" or unlocking coherent communications at 2 microns.

Response:

We thank the reviewer for this constructive suggestion. As suggested, we have rephrased every occurrence of “unconventional bands,” in our revised manuscript and we have changed the title to “Super-broadband On-chip Continuous Spectral Translation Unlocking Coherent Optical Communications Beyond Conventional Telecom Bands”

Reviewer’s comment 2:

Line 21: replace “This enables unprecedented coherent transmission in the 2- μm band using well-developed conventional C-band transmitters and coherent receivers” with “this paper/letter shows a 700nm [or THz, whatever is the best number] spectral translator for coherent transmission in the 2- μm band using well-developed conventional C-band transmitters and coherent receivers, as an example of this device potential, which could also unlock communications at other wavebands.”

Response:

We thank the reviewer for the suggestion of clarifying the potential of the spectral translators. We have replaced the sentence with

“We demonstrate coherent transmission in the 2- μm band using well-developed conventional C-band transmitters and coherent receivers, as an example of the potential of the spectral translators that could also unlock communications at other wavelength bands.”

Reviewer’s comment 3:

Lines 26-39 are not relevant, and not the best motivation for the technology. Better to start at line 39 at “The potential to open up...”

Response:

We would like to keep these descriptions of coherent communications as a background introduction, as our demonstration shows coherent transmission at the 2- μm band. This could help explain why we prefer coherent transmission in the 2- μm band (spectral efficiency), and why previous demonstrations are focused on the IM/DD systems (due to the lack of coherent transceivers).

Reviewer’s comment 4:

Then line 44: I would replace with “However, it is generally very challenging to build a complete coherent optical communication system for wavelength bands away from C+L, due to the lack of

telecom-grade, commercially available narrow-linewidth lasers, I/Q modulators, and coherent receivers, including for bands such as O-, E-, S- or U-, and beyond.”

Response:

We think the reviewer’s suggestion better describes the new wavelength bands than “unconventional bands.” We have modified our sentence to

“However, it is generally very challenging to build a complete coherent optical communication system for wavelength bands away from C and L, due to the lack of telecom-grade, commercially available narrow-linewidth lasers, I/Q modulators, and coherent receivers, including for wavelength bands such as O, E, S or U, and beyond.”

Reviewer’s comment 5:

Line 50, replace: “An alternative solution is the use spectral translation technology to provide a bridge between mature telecom bands and unconventional wavelength bands” with “An alternative solution is the use of spectral translation [reference] as a means to provide a bridge between telecom bands to any other wavelength band, including 2microns in this case.”

Response:

We appreciate the reviewer’s helpful suggestion to avoid the vague term “unconventional wavelength bands”. We have changed the sentence to:

“An alternative solution is the use of spectral translation as a method to provide a bridge between the telecom bands and any other wavelength band of interest for optical transmission.”

Reviewer’s comment 6:

Line 59 onwards – very repetitive, including the paragraph that follows it. Replace with “However, the nature of the resonance gives a discrete translation band in frequency and a narrow resonance bandwidth, which limits the data rate and compromises the potential on opening up the new wavelength bandwidth.” I would also delete “Aiming for coherent optical communication with high speed and high spectral efficiency in unconventional wavelength bands, continuous (in both time and frequency) spectral translation is necessary but has not yet been demonstrated”.

Response:

We thank the reviewer for pointing out some redundancy in the text. We have changed the text from line 59 and the first sentence of the next paragraph to

“However, the nature of the resonance gives a discrete translation band in frequency and a narrow resonance bandwidth, which limits the data rate and compromises the potential of opening up the new wavelength bandwidth.”

“Here, we propose and demonstrate continuous (in both time and frequency) spectral translation for coherent optical communication in the 2- μm band as an example of the potential to unlock new wavelength bands utilizing conventional C-band transceivers.”

Reviewer’s comment 7:

Line 65, I would replace “Here, we propose and demonstrate a continuous spectral translation scheme for coherent optical communication at 2microns, as an example of the potential for unlocking the use of new wavelength windows utilizing conventional C-band transceivers. ”

Response:

As addressed in the previous comment #6, we have changed the sentence to

“Here, we propose and demonstrate continuous (in both time and frequency) spectral translation for coherent optical communication at the 2- μm band as an example of the potential to unlock new wavelength bands utilizing conventional C-band transceivers.”

Reviewer’s comment 8:

Line 68 - Delete “We introduce the basic principle of continuous spectral translation and our AlGaAsOI platform for coherent optical communication in unconventional wavelength bands. We target the 2- μm wavelength band for the demonstration”. It’s not well written, it’s repetitive, and the previous sentences already explain that.

Response:

We thank the reviewer for pointing out the redundant text and we have deleted these sentences.

Reviewer’s comment 9:

Line 72 onwards, replace with “With the fabricated AlGaAsOI nanowaveguides as spectral translators, as well as a short-wavelength thulium-doped fibre amplifier (TDFAs) and hollow-core fibre (HCF), we have enabled coherent transmission in the 2- μm band using C-band transmitters and coherent receivers. We have successfully transmitted 4 \times 32-Gbaud 16-QAM Nyquist wavelength-division multiplexing (N-WDM) signals over a 1.15-km HCF at the 2- μm wavelength band with a spectral efficiency of 2.4 bit s⁻¹ Hz⁻¹”. The words “in-house”, or “unprecedented” are not appropriate. The matter of fact is how wide your spectral translation is, and everything else

is an application of the device. TDFA and HCF developed by ORC that were used here were demonstrated before, and well referenced in this paper. Focus on what this paper is really trying to say. Perhaps here you should focus on operations of the spectral translator, “cw pumped at xxx, with yy bias, and at zzz oC” or whatever the actual operations details were – this gives a guide to the audience on the potential comparison with the nonlinear micro-ring resonators you mentioned before.

Response:

We thank the reviewer for the help to clarify the focus of this paper. We have removed “unconventional bands” and changed the text from line 72 to:

“With the fabricated AlGaAsOI nanowaveguides as spectral translators, as well as a short-wavelength thulium-doped fibre amplifier (TDFA) and hollow-core fibre (HCF), we have enabled coherent transmission in the 2- μm band using C-band transmitters and coherent receivers. We have successfully transmitted 4 \times 32-Gbaud 16 quadrature-amplitude modulation (QAM) Nyquist wavelength-division multiplexing (N-WDM) signals over a 1.15-km HCF at the 2- μm wavelength band with a spectral efficiency of 2.4 bit s⁻¹ Hz⁻¹.”

Reviewer’s comment 10:

Line 79: the end is calling for the addition of other applications beyond comms, perhaps space applications, sensing, astronomy?! This draws further audiences to read your paper, and Nature-based publications tend to attract a wider community!

Response:

We thank the reviewer for the reminder of listing possible new applications. We actually have stated several possibilities in our cover letter:

“We believe that this work could resonate well not only within the optical communication research community, but also with a larger scientific crowd in broader applications such as massive photonic integration in two-photon absorption (TPA)-free wavelength bands, high-resolution spectroscopy for uncharted wavelength bands, as well as high-sensitivity infrared astronomy (like NASA’s Two Micron All Sky Survey (2MASS)), due to the provided possibility of processing signals at wavelength bands that are currently inaccessible using current signal processing devices. Therefore, we believe this manuscript would be very valuable for the broad audience of Science.”

We have now included them here in the paper:

“Beyond the application in optical transmission, the spectral translators may also find significance in a wider range of applications such as massive photonic integration in TPA-free wavelength

bands, high-resolution spectroscopy for uncharted wavelength bands, as well as high-sensitivity infrared astronomy, due to the opportunities they provide for processing signals in wavelength bands that are currently inaccessible with today's devices.”

Reviewer's comment 11:

Lines 103-109 are excellent. I do like the new sentences there.

Response:

We appreciate the comment from the reviewer.

Reviewer's comment 12:

Line 109 onwards I would re-write as “Optical communication in the 2- μ m band, to the best of our knowledge, has focused on intensity-modulation and direct-detection [refs]. Enabling a 2- μ m-band coherent optical communication system based on spectral translation requires a sufficiently powerful 1.74- μ m pump and AlGaAsOI nanowaveguides with a conversion bandwidth >450 nm. For the 1.74- μ m pump, we have built an all-fibre TDFA working in the 1700-1800 nm band, achieving an output power of >500 mW [ref]. The design and fabrication details of the short-wavelength TDFA are given in the Methods.”

Response:

As suggested, the text has been modified to

“Optical communication in the 2- μ m band, to the best of our knowledge, has focused on intensity modulation and direct detection. Enabling a 2- μ m-band coherent optical communication system based on spectral translation requires a sufficiently powerful 1.74- μ m pump and AlGaAsOI nanowaveguides with a conversion bandwidth >450 nm. For the 1.74- μ m pump, we have built an all-fibre TDFA working in the 1700--1800 nm band, achieving an output power of >500 mW. The design and fabrication details of the short-wavelength TDFA are given in the Methods.”

Reviewer's comment 13:

Line 116 – I still can't see the fabrication details, dimensions and SEM picture of the AlGaAsOI nanowaveguide. Figure 1 has a representative picture, but it needs a new picture with the design (cross-section of the different layers / dimensions / compositions / refractive indexes), length etc. Furthermore, were tapers used? Or grating coupler? How was it coupled to fibre? Loss? It says that this information is in “Supplemental Information #3”, but I can't find it in the reviewer pack. I can only see on “Methods”, and the figure is still missing.

Response:

We thank the reviewer for the suggestion of adding an SEM picture of the AlGaAsOI nanowaveguide. The Methods section does not allow figures. We have put a new SEM image in Fig. 1. This AlGaAsOI nanowaveguide in the SEM image does not have the exact same dimensions as the ones used in the experiment. Therefore, we would like not to show its real dimensions in Fig. 1, but only as an illustration of the cross-section. The detailed dimensions are stated in the Methods. Other details such as fabrication, dimensions, and coupling, have already been included in the Methods.

Figure R2-1 | Measured insertion losses of a specialized set of nine AlGaAsOI nanowaveguides, each with a cross-section dimension of 470×350 nm and input/output inverse tapers of the same width. This indicates the difference in coupling and waveguide losses for the C-band and the 2- μ m-band signals.

The insertion loss of the AlGaAsOI nanowaveguide consists of the loss of the AlGaAsOI nanowaveguide and the loss between the input/output inverse taper and the tapered fiber.

We have fabricated a specialized set of nine AlGaAsOI nanowaveguides with a cross-section dimension of 470×350 nm. Each AlGaAsOI nanowaveguide has a pair of input/output inverse tapers of the same width. Figure R2-1 shows the measured insertion losses for the C-band signal and the 2- μ m-band signal. **This shows the difference in coupling and waveguide losses for the C-band and the 2- μ m-band signals.** However, we could not get the exact numbers for the loss of the AlGaAsOI nanowaveguide itself and the inverse tapers. This is because if we use the cutoff method to derive the waveguide loss, we have to bend the waveguide, which would generate higher modes.

We have fabricated many of the AlGaAsOI nanowaveguides with a cross-section dimension of 910×350 nm with a 5-mm length. In the experiment, we select the waveguides so that the

conversion efficiency is maximized. The result is that the AlGaAsOI nanowaveguide 1 has an input inverse taper of 140 nm and an output inverse taper of 180 nm to maximize the output of the 2- μ m-band signal, and the AlGaAsOI nanowaveguide 2 has an input inverse taper of 150 nm and an output inverse taper of 130 nm to maximize the output of the C-band signal. **The measured insertion loss for the AlGaAsOI nanowaveguide 1 is 13.49 dB for the C-band signal and 19.13 dB for the 2- μ m-band signal.**

However, since we did not have a test set of AlGaAsOI nanowaveguides with a cross-section dimension of 910 \times 350 nm, we can not tell whether the waveguide loss or the loss of the inverse tapers dominates. In addition, conversion efficiency is more important for the application. Therefore, we did not include the loss of the AlGaAsOI nanowaveguides in our manuscript. The reference for the loss of the AlGaAsOI nanowaveguide can be found here:

[R2-1] Ottaviano, L., Pu, M., Semenova, E. & Yvind, K. Low-loss high-confinement waveguides and microring resonators in AlGaAs-on-insulator. *Opt Lett* 41, 3996–3999 (2016).

[R2-2] Pu, M. et al. Ultra - Efficient and Broadband Nonlinear AlGaAs - on - Insulator Chip for Low - Power Optical Signal Processing. *Laser Photonics Rev* 12, 1800111 (2018).

Reviewer's comment 14:

Line 131 – this is the octave claim. Is this true for the same device? Or would different structures/composition be required to achieve the octave spectral translation?

Response:

This is a very good question. The octave translation bandwidth is claimed from the simulation of the device, with a cross-section dimension of 910 \times 350 nm and pump wavelength of 1739.8 nm. We can achieve an even larger translation bandwidth with other waveguide dimensions. The octave translation bandwidth shows that this spectral translation scheme has great potential in opening up many new wavelength bands.

In the experiment, unfortunately, we can not measure the whole translation band, due to the lack of a widely tunable laser. A tunable laser working from 1200 nm to 1750 nm is preferred. We do have measured the translation band from 1500 nm to 1580 nm, and the result is shown in Supplementary Note 3. We have also generated an optical frequency comb in the C band and translated it into the 2- μ m band to confirm the flatness of the translation band. The result of the translation of an optical frequency comb is shown in Supplementary Note 4.

We have added the wavelength range that we measured in the revised paper:

“The measured flatness of the translation band from 1500—1580 nm can be found in Supplementary Note 3.”

Reviewer’s comment 15:

Line 141 – rephrasing to “The 1.55-um-band transmitter and receiver comprised of discrete components in this demonstration for flexibility, but commercial transceivers would equivalently work with the spectral translators”. Remove “without obstacles”, as it’s a fact, otherwise you need to explain what obstacles you were thinking about. So just delete it.

Response:

As suggested, the term “without obstacles” is now deleted.

Reviewer’s comment 16:

Line 145 – rephrasing to “A CW single mode [was it single mode?] distributed feedback (DFB) laser was used as a pump, thermally tuned to a central wavelength of 1739.74 nm with a linewidth of 2 MHz and a power of xx dBm. In order to simultaneously optically pump each nanowaveguide device (one at the transmitter and one at the receiver), and to maximize the CE, the CW pump was amplified by a short-wavelength TDFA and split into two with a 50/50 coupler (as it 50/50?), with measured optical powers of 20.0 dBm and 19.9 dBm just before launching to the input tapered couplers of each of the AlGaAsOI nanowaveguides (1 and 2), respectively.” Note: were the nanowaveguides fibre coupled? Or was it launched via free-space?

Response:

We thank the reviewer for the suggestion to make the description clear. The AlGaAsOI nanowaveguides are coupled using tapered fibres. On the AlGaAsOI chip, there are a pair of tapered couplers placed at each side of the straight waveguide and optimized for low coupling losses.

We have changed the description to

“A CW single mode distributed feedback (DFB) laser is used as a pump, thermally tuned to a central wavelength of 1739.74 nm with a linewidth of 2 MHz and a power of 5 dBm. In order to simultaneously pump each nanowaveguide device (one at the transmitter and one at the receiver), and to maximize the CE, the CW pump was amplified by the short-wavelength TDFA and split into two with a 3-dB coupler, with measured optical powers of 20.0 dBm and 19.9 dBm just before the input tapered couplers of AlGaAsOI nanowaveguides 1 and 2, respectively.”

Reviewer’s comment 17:

Figure 4: I must say that the colour code is unfortunate and the contrast is poor. I would imagine that people who have colour-reading issues would find difficulty in understanding the results. Always have in mind what it would look like if the figure was black and white, can you still see differences? My suggestion is to use very clear symbols (open/close) for pre- / post- compensation, use larger symbols etc. Hopefully the editors will work with you on them. Also confusing is the constellation diagrams. For what power level were they taken? An arrow to show that would be good.

Response:

We thank the reviewer for the suggestions. We tried to use limited colours for the whole manuscript so that it does not feel too busy. We have implemented different symbols to differentiate these BER curves in the revised manuscript. We have also stated in the caption where the constellation diagrams are acquired:

“Insets give the typical constellation diagrams of the 16-QAM signals that can achieve error-free performance after LDPC decoding.”

Figure 4 | Bit error ratio (BER) measurement for the 2- μ m-band signals using the 2- μ m-band transmitter and receiver based on the spectral translators. a, BER performance of the single-channel 2- μ m-band 16-QAM signal at back-to-back. **b,** BER performance of the single-channel 2- μ m-band 16-QAM signal after 1.15-km HCF transmission. Error-free performance is achieved for both back-to-back and transmission scenarios with 20%, 25%, and 33% LDPC overheads, indicating a good performance margin. **c,** BER performance of the 2- μ m-band 4 \times 32 Gbaud 16-QAM signal at back-to-back. **d,** BER performance of the 2- μ m-band 4 \times 32 Gbaud 16-QAM signal after 1.15-km HCF transmission. With 33% LDPC overhead, error-free performance is achieved for both back-to-back and transmission scenarios. Insets give the typical constellation diagrams of the 16-QAM signals that can achieve error-free performance after forward error correction (FEC) decoding.

Reviewer's comment 18:

Line 199 onwards (from “with a cutting-edge” etc. These are subjective wording, the use of plain English is paramount) - rephrase it to “With a short-wavelength TDFA and HCF, we have demonstrated the first spectrally translated 2- μ m-band coherent optical transmission of high capacity N-WDM signals with a spectral efficiency of 2.4 bit/s/Hz. The spectral translation scheme could be adapted to other wavelength bands, since the AlGaAsOI nanowaveguides can unlock at least an octave spanning conversion bandwidth. This approach could facilitate the use of new wavelength bands for coherent optical communications still utilizing conventional C-band

transceivers and/or transponders. We also believe this scheme has significance in broader applications beyond optical communications, such as in massive photonic integration in TPA-free wavelength bands and in high-resolution multi-banded spectroscopy, through the possibility of coherently transmitting, detecting, and processing signals at the wavelength bands that are currently inaccessible using current signal processing devices.”

Response:

We appreciate the reviewer’s suggestion. We have now rephrased the paragraph into:

“With a short-wavelength TDFA and HCF, we have demonstrated the spectrally translated 2- μm -band coherent optical transmission of high-capacity N-WDM signals with a spectral efficiency of $2.4 \text{ bit s}^{-1} \text{ Hz}^{-1}$. The spectral translation scheme could be adapted to other wavelength bands, since the AlGaAsOI nanowaveguides can unlock at least an octave spanning conversion bandwidth. This approach could facilitate the use of new wavelength bands for coherent optical communications still utilizing conventional C-band transceivers. We also believe this scheme has significance in broader applications beyond optical communications, such as in massive photonic integration in TPA-free wavelength bands, in high-resolution multi-banded spectroscopy, and in high-sensitivity infrared astronomy, through the possibility of coherently transmitting, detecting, and processing signals at wavelength bands that are currently inaccessible using current signal processing devices.”

Reviewer’s comment 19:

Line 210 - The design and fabrication of the AlGaAsOI nanowaveguide: It needs a picture with the details in here. See comment #13. I cannot see “supplemental information #3” in the reviewer pack, so I cannot comment on that.

Response:

As we have mentioned in the response to the reviewer’s comment #13, we have put the SEM image in Fig. 1. This AlGaAsOI nanowaveguide in the SEM image does not have the exact same dimensions as the ones used in the experiment. Therefore, we would like not to show its real dimensions in Fig. 1, but only as an illustration of the cross-section. The detailed dimensions are stated in the Methods.

Reviewer #2 (Remarks to the Author):

The Authors fully addressed my remarks that are properly clarified within the revised article. Therefore, my recommendation is for the article acceptance for publication.

Response:

We thank the reviewer again for the comments and suggestions on the previous round of modifications.